# Barriers to childhood asthma care in sub-Saharan Africa: a multicountry qualitative study with children and their caregivers

Kimesh Loganathan Naidoo [1,2] Sindisiwe Dladla [1]
Reratilwe Ephenia Mphahlele [1] Gioia Mosler [3] Sophie Muyemayema [4]
Andrew Sentoogo Ssemata [5] Elizabeth Mkutumula [6]
Olayinka Olufunke Adeyeye [7] Melinda Moyo [4] Olayinka Goodman [7]
Yetunde Kuyinu [7] Rebecca Nantanda [8,9] Ismail Ticklay [4,10]
Hilda Angela Mujuru [4] Jonathan Grigg [3] Refiloe Masekela [1]

JG and RM are joint senior authors.

For numbered affiliations see end of article.

**Correspondence to**
Dr Kimesh Loganathan Naidoo;
naidook9@ukzn.ac.za

## ABSTRACT

**Objectives** This study identifies barriers and provides recommendations to improve asthma care in children across sub-Saharan Africa, where qualitative data is lacking despite high rates.

**Design** One of the aims of our National Institute for Health Research global health research group 'Achieving Control of Asthma in Children in Africa' was to use qualitative thematic analysis of transcribed audio recordings from focus group discussions (FGDs) to describe barriers to achieving good asthma control.

**Setting** Schools in Blantyre (Malawi), Lagos (Nigeria), Durban (South Africa), Kampala (Uganda) and Harare (Zimbabwe).

**Participants** Children (n=136), 12–14 years with either asthma symptoms or a diagnosis and their caregivers participated in 39 FGDs. All were recruited using asthma control questions from the Global Asthma Network survey.

**Results** There were four key themes identified: (1) Poor understanding, (2) difficulties experienced with being diagnosed, (3) challenges with caring for children experiencing an acute asthma episode and (4) suboptimal uptake and use of prescribed medicines. An inadequate understanding of environmental triggers, a hesitancy in using metred dose inhalers and a preference for oral and alternate medications were identified as barriers. In addition, limited access to healthcare with delays in diagnosis and an inability to cope with expected lifestyle changes was reported. Based on these findings, we recommend tailored education to promote access to and acceptance of metred dose inhalers, including advocating for access to a single therapeutic, preventative and treatment option. Furthermore, healthcare systems should have simpler diagnostic pathways and easier emergency access for asthma.

**Conclusions** In a continent with rapidly increasing levels of poorly controlled asthma, we identified multiple barriers to achieving good asthma control along the trajectory of care. Exploration of these barriers reveals several generalisable recommendations that should modify

asthma care plans and potentially transform asthma care in Africa.

**Trial registration number** 269211.

## STRENGTHS AND LIMITATIONS OF THIS STUDY

⇒ This is the first multicountry African qualitative study that used thematic analysis to identify barriers to asthma care.
⇒ Perspectives of both adolescents and their care-givers on asthma diagnosis and management are included.
⇒ Views of both children with a physician diagnosis of asthma and those with symptoms of asthma and not formally diagnosed are included.

## INTRODUCTION

Globally, asthma is the most common non-communicable respiratory disease among children.[1] While the prevalence of asthma in African children is currently 14% and matches the global average, it is increasing in low-middle income countries with different trajectories suggested by the wide range of prevalence between countries, for example, Kenya at 15.8%, Nigeria 13.0% and South Africa 20.3%.[1–4] The goal of asthma care is to optimise symptom control and maintain a good quality of life.[5 6] Asthma control has been reported to be poor with the 2018 Global Asthma Report reporting high uncontrolled asthma prevalence rates across sub-Saharan Africa (sSA).[4] The determinants of poor asthma control in sSA are multifactorial including environmental factors which are certainly important and include rural to urban migration, early life urban residence, overcrowding, poorer air quality, lifestyle

changes and diet.[6–9] Patient-related factors reported in African children include poor knowledge and symptom recognition, non-acceptance of inhaler treatment, older age and concurrent allergic rhinitis.[6 7] Health system factors include limited diagnosis of asthma, episodic management of asthma symptoms at the primary healthcare (PHC) level and poor access to inhaled corticosteroids.[5 6 9–12] However, to date, there is little qualitative data from children and young people on the determinants of asthma control in this region. Studies in high-income countries have found that children's and caregivers' health beliefs about asthma and poor relationships with healthcare providers (HCPs) can undermine asthma management.[13] Caregivers have also reported fear, feeling of powerlessness, uncertainty and social tension related to their children who have asthma.[14–16] Studies from individual African countries have also identified physical, emotional and social burdens related to childhood asthma, and these have been postulated to affect nonadherence to recommended treatment regimens.[10 17–19]

Understanding the views of caregivers and affected children with doctor-diagnosed asthma and those with symptoms suggestive of asthma are needed to capture perspectives effectively. To accurately and efficiently determine strategies to improve asthma care, recommendations need to be identified and explored, that are derived from the perspectives of affected African children and their caregiver's concerns.[20] A richer, qualitative understanding of identified barriers to asthma care in African countries will assist with the development of relevant advocacy and policy interventions across the continent.[21] This study aimed to capture the views of children with asthma/asthma symptoms with and their caregivers, in sSA and to identify barriers to asthma diagnosis and care as well as interventions to improve asthma care across countries in Africa.

## METHODS
### Study design
This study was a component of the achieving control of the National Institute for Health Research funded global health research group's 'Achieving control of asthma in children in Africa' (ACACIA), a larger observational cohort study across six African countries; Ghana, Malawi, Nigeria, South Africa, Uganda and Zimbabwe.[22] The ACACIA project aims to determine the burden of asthma in school-going children by assessing asthma control in children between 12 and 14 years with symptoms compatible with asthma or a doctor-diagnosis of asthma to assess symptom severity and asthma control. As a separate study, we conducted focus group discussions (FGDs) with children with an asthma diagnosis or symptoms suggestive of asthma and their caregivers to obtain their perspectives on the barriers to managing asthma. The study plan, procedures and methods were standardised across all countries.[22]

### Study population
Children and their caregivers were recruited initially as part of the ACACIA data set. A Breathing Survey developed from questions in the Global Asthma Network questionnaire (online supplemental file 1) was used to identify participants between 12 and 14 years with asthma symptoms or an asthma diagnosis.[22] Focus group participants (both children and their caregivers) were recruited using purposive sampling and categorised into four groups; children with a doctor-diagnosis of asthma (CAD) and their parents/caregivers (PAD), children with severe wheezing symptoms without a doctor-diagnosis of asthma (CND) and their parents/caregivers (PND), with FGDs held separately for each category. The aim of this group categorisation was to ensure views of both children and caregivers, as well as those with and without a formal asthma diagnosis, were adequately sampled.

### Study setting
The FGDs were held between 1 November 2020 and 30 June 2021 in schools in cities across countries: Blantyre, (Malawi); Lagos, (Nigeria); Durban, (South Africa); Kampala, (Uganda); and Harare, (Zimbabwe). Data from Ghana were not included for analysis due to challenges with COVID-19 restrictions in that country that impacted data collection. The in-country team of researchers and fieldworkers facilitated the FGDs.

### Data collection
Focus groups were conducted face-to-face, with adherence to in-country-specific COVID-19 regulations with at least 1–2 FGD/s per category per country. All FGDs were moderated by at least two facilitators, one of whom was a study investigator at each site. FGDs took place in either English and/or isiZulu, in South Africa (SA), Shona (Zimbabwe) or Chichewa (Malawi). Other centres only used English. Focus groups were audio-recorded, transcribed verbatim and translated to English if needed in the country. Discussions lasted between 40 and 60 min and followed a predefined question guide specific to each focus group category. The discussion guides were developed after multiple iterations by the ACACIA core study team and informed by the study aims and literature review on barriers to asthma care (online supplemental file 2).

### Data analysis
The transcripts taken during the discussion were used to analyse the data using the NVivo V.1.6 software program. We used an inductive analysis to examine the data.[23] Ten members of the ACACIA research group from all the participating countries independently developed codes from the transcripts and then discussed the coding to develop a code book.[23 24] At least two investigators reviewed each transcript. A reiterative process was followed to apply the codebook to the transcribed data and determine whether further codes emerged. When no new codes emerged, the codebook was assumed to be

a valid representation of the data and finalised (online supplemental file 3).

Braun and Clarke's six-step thematic analysis was then used as the analytical framework to identify the patterns and themes within the coded data.[25] Major themes and subthemes were identified based on those commonly occurring codes in all countries. These subthemes were highlighted and grouped into major themes based on commonly occurring patterns and relationships noted between the identified codes. All investigators reached consensus on these major themes and grouping of subthemes after multiple iterations. Quotes chosen to illustrate the themes derived from the data were specifically selected only if they resonated across all countries.

### Patient and public involvement

The focus groups were part of the wider ACACIA study and its patient and public involvement.[22] The focus groups set-up, especially for children, was developed at each site with the local school staff, to provide a friendly and safe experience in a familiar environment. We tested the questions used in the focus group discussion at one of our sites (Nigeria) before other sites. Patients were involved at several stages of the ACACIA study, but not directly for the recruitment and conduct of the focus group element. A summary of all ACACIA study results will be disseminated to the participating schools, who will communicate it to students and parents

Data was de-identified and stored on a password protected computer with access only to researchers.

## RESULTS

### Description of the focus groups conducted

The data from 39 transcribed FGDs reflecting 256 participants across five countries were analysed. South Africa held 15 FGDs compared with 4–6 in all other countries. This was due to more parents and children who indicated willingness to participate in FDGs in SA during the COVID-19 pandemic than all other countries. Table 1 indicates the breakdown of these groups per category and country. A total of 136 children were included with CAD 45.6% (n=62) and CND 54.4% (n=74). Of the 120 caregivers, they included PAD 42.5% (n=51) and PND 57.5% (n=69).

### Summary of findings

Table 2 lists the subthemes and themes identified and figure 1 shows the relationships. There were four major themes identified which were grouped into: (1) Poor understanding of asthma, (2) difficulties experienced with being diagnosed with asthma, (3) challenges with caring for children experiencing an acute asthma episode and (4) suboptimal uptake and use of prescribed asthma medicines.

### Poor understanding of asthma

This theme related to perspectives on how caregivers and children defined asthma and what was considered the cause of asthma. We could not ascertain differences in knowledge among caregivers of those with and without a diagnosis of asthma. We did find a distinction in how caregivers and children understood the causation of asthma. Children expressed fewer opinions on the familial or

**Table 1** Summary of children with asthma or asthma symptoms and their caregivers who participated in focus group discussions (n=256)

| | Children n=136 | | | | Caregivers n=120 | | | |
|---|---|---|---|---|---|---|---|---|
| | Children with a doctor-diagnosis of asthma (CAD) | | Children with severe asthma symptoms but no diagnosis (CND) | | Caregivers of children with a doctor-diagnosis of asthma (PAD) | | Caregivers of children with severe asthma symptoms but no diagnosis (PND) | |
| Country (n=total number of focus groups) | Focus group (n) | Number of participants | Focus group (n) | Number of participants | Focus group (n) | Number of participants | Focus group (n) | Number of participants |
| Malawi (n=4) | 2 | 8 | 1 | 8 | 1 | 6 | 0 | 0 |
| Nigeria (n=7) | 2 | 14 | 2 | 12 | 2 | 12 | 1 | 8 |
| South Africa (n=15) | 3 | 19 | 3 | 29 | 3 | 21 | 6 | 41 |
| Uganda (n=7) | 2 | 12 | 1 | 6 | 2 | 12 | 2 | 19 |
| Zimbabwe (n=6) | 1 | 9 | 3 | 19 | 0 | 0 | 2 | 10 |
| Total participants (n=256) | **10** | **62** | **10** | **74** | **8** | **51** | **11** | **69** |

**Table 2** Major themes and subthemes and their relationships

| Major themes | Poor understanding of asthma | Difficulties experienced with being diagnosed with asthma | Challenges with caring for children experiencing an acute asthma episode | Suboptimal uptake and use of prescribed asthma medicines |
|---|---|---|---|---|
| Subthemes | ▶ Confusion about asthma definitions.<br>▶ Lack of clarity about what causes asthma. | ▶ Concerns about delays in getting an asthma diagnosis.<br>▶ Disruptions in child's life on being diagnosed with asthma.<br>▶ Negative responses from significant others to child's asthma diagnosis. | ▶ Confusion around triggers of an acute asthma episode.<br>▶ Lack of knowledge on recognising and responding to an acute asthma episode.<br>▶ Difficulties in access to hospital care when a child experiences an acute asthma episode. | ▶ Preferences for trying alternative/complementary treatments.<br>▶ Poor acceptance with using inhalers. |

genetic nature of asthma than their caregivers. In all groups, there was a poor understanding of how environmental triggers and other factors such as prematurity and genetic predisposition contribute to the development of asthma.

> There is dirt of some kind that interrupts ones breathing in their lungs so when air is meant to come in then you'll find that it's this dirt that is blocking the inhalation and exhalation of air. (PND South Africa (ZA), Group 6)

### Difficulties experienced with being diagnosed with asthma

This theme related to perspectives being expressed about delays in being diagnosed, reflected predominantly in the groups with a formal doctor-diagnosis. Specific concerns

related to a perceived reluctance of healthcare workers in making an asthma diagnosis with resulting frustration these delays cause.

> I think that symptoms start with cough and flu. We waste a lot of time and yet the actual disease is not treated. So, after you have given that dose of cough and flu, when you go back, the doctor will tell you that is an infection yet it is asthma. (PAD, Uganda (UG), Group 2)

In addition, this theme reflected on the emotional distress felt with disruptions to lifestyles due to acute asthmatic episodes. Both children and caregivers indicated displeasure about the disruption in activities of normal daily life after being diagnosed with asthma. These included disruptions to school attendance, participation in sports and the child's share of household chores. Children expressed emotions of sadness and loss while caregivers expressed anxiety related to the disruption of their previous activities.

> When am in an attack I can't do anything, I can't study, I can't go to class, I can't breathe properly, I can't focus, I can't understand. I can't be at school. (CAD, Malawi (MW), Group 1)

> Since we discovered she has asthma, my daughter is not exposed to a lot of things anymore. We don't allow her to sweep except if she's covered with a face mask before sweeping, we stopped her from drinking cold water and her room is cleaned properly, no rug, no cockroaches in the house, the fan is kept clean. (PAD, Nigeria, Group 2)

Children also specifically expressed concerns about the stigma and discrimination they endured when their asthma diagnosis was known in the school environment. These reactions from significant others impacted negatively on children and caregivers leading to feelings of isolation and this promoted non-disclosure

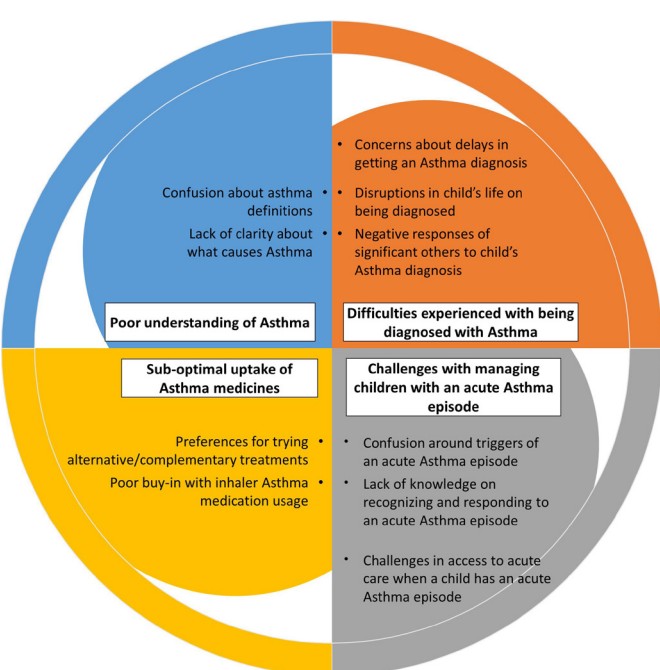

**Figure 1** Relationships of the major themes and subthemes identified.

It's also embarrassing because they all look at you funny and some start excluding you from games because they fear that you will have an attack. Some end up avoiding you forever and you remain with no one to play with. (CAD, ZW, Group 1)

I don't tell people that I am asthmatic because they will start looking at me all different. (CAD, UG, Group 2)

### Challenges with caring for children experiencing an acute asthma episode

In all groups, identifying and avoiding various asthma triggers was seen as an important component in managing symptoms.

Simple solutions like make sure your surroundings are clean, check for dampness, in your bathroom or the ceilings and your walls … there are lots of things you can do to ensure that, the triggers are reduced. (PND, ZA, Group 2)

For caregivers, these avoidance strategies translated to altering the external environments of their children, while children focused more on how lifestyle changes impacted their daily lives. All groups articulated that they had difficulties with understanding what the various environmental triggers are and how to avoid them effectively.

I was outside and then my sister started sweeping the compound raising dust, the dust affected me and I started wheezing. I was then told to go inside the house and sleep. (CAD, Zimbabwe (ZW), Group 1)

The theme also includes caregivers' perspectives on their interactions the health systems, with challenges in accessing acute care when a child experiences an acute asthma episode.

In these interactions, caregivers reflected on their frustrations, financial dependencies and vulnerabilities. These arose following the experiences of limited and often difficult access to healthcare during an acute asthmatic attack.

Government hospitals are useless … It is possible that the doctor's house is very near but for him to come is a challenge. Sometimes they will leave in the middle of their shift and you come with your child to the hospital and they tell you to wait … meanwhile your child's condition is worsening. (PAD, MW, Group 1)

Apart from the hospital, I would also seek help from more accessible nurses in the neighbourhood or at home because these attacks usually happen when the hospital is too far and, in that moment, they may not be able to help you. (PND, UG, Group 1)

### Suboptimal uptake and use of prescribed asthma medicines

Across all groups, common perspectives on the role, availability and acceptance of various types of medications to treat asthma were identified. All groups reflected on relatively easier access to and acceptance of alternate and complementary asthma treatments than prescribed asthma medications.

Others told me to find a chameleon and leave it on flour for it to walk over, then to use the flour to make porridge to give the child. None of those things really worked! I have taken too much of these traditional medications! (PAD, MW, Group 1)

I also use the inhaling smoke method. My mother does the same thing of putting marijuana seeds on hot coal and covers me with a blanket to inhale the smoke but it will be for five minutes tops and it clears my chest and I breathe better. (CND, ZW, Group 1)

Juxtaposed to the acceptance of various alternate medications and treatment, there are widely held views that using and relying on inhalers should be met with caution. In addition, caregivers and children with asthma and no asthma diagnosis expressed various fears and myths about using prescribed inhalers. Concerns over possible addiction developing with the use of inhalers and fears of what could happen when inhalers are unavailable were expressed. In addition to these concerns, there is a clearly articulated preference for oral over inhaled medication by many.

Okay a lot of people would tell us …we can use the inhalers but when I grow up it will build up in my body … also what if I get addicted to the inhaler and one day I have shortness of breath and don't have the inhaler? What will happen? (CAD, MW, Group 2)

The people from my village saw that I am using an inhaler, they asked me why I have started using it as if I don't know how such things end. But since last year when I started using the inhaler, everything is fine and we have not been admitted in the hospital since. I really had that mind-set that inhalers kill but since I started using them I saw that they are really helpful. (PAD, MW, Group 1)

### Barriers identified across countries

From the themes, we identified several barriers to asthma care across the five African countries, box 1. Although there was some understanding of the role of genetic predisposition and allergen triggers, children and caregivers still expressed the need for more information, specifically regarding trigger avoidance. This included lifestyle changes that had to be made for families and disruption of normal family routines and chores. Reliance on the use of oral and complementary medications was expressed with the stigma around the use of asthma inhaler therapies.

### DISCUSSION

Several barriers to asthma care that were common in all five African countries in this study were identified. The relationships of these barriers to optimal asthma care in Africa is highlighted specifically from the participants'

## Box 1    Identified barriers to asthma care across African countries

Barriers to asthma care across the African countries
⇒ Inadequate understanding of environmental triggers and causes of asthma impact on optimal, holistic asthma care.
⇒ Reluctance in healthcare workers to make an asthma diagnosis results in delays in diagnosing asthma.
⇒ Difficulties experienced coping with adjustments in lifestyles required with asthma hamper prevention of acute episodes.
⇒ Limited access to healthcare during an acute asthmatic episode.
⇒ Preferred usage of alternate and complementary asthma treatments delays uptake of prescribed asthma medication.
⇒ Preferences in using oral asthma medication delays early adoption of inhaler medication.
⇒ Fears and myths prevent uptake use of metred dose inhalers.

perspectives. On further exploration, several cross-cutting recommendations were extrapolated from the analysis of these relationships of the barriers to caring for children with asthma and these were derived from the author's perspectives.

### Inadequate understanding of environmental triggers and genetic predisposition in causing asthma

Asthma is a complex heterogeneous condition with multiple risk factors. The role of genetics, atopy and environment is complex even for experienced researchers. It is therefore not surprising that there was a poor societal understanding of the differences between cure, prevention and control, including confusion between trigger avoidance and symptom control using chronic treatments. Studies from sSA have noted that caregivers and children ascribe triggers with causation.[10 17] Thus, removing these 'offenders' equates to treatment/cure, lessening the perceived necessity for measures to prevent or control asthma.[10 17] These perspectives are identified in our study's findings of an inadequate understanding of environmental triggers, such as dust and cold weather. We speculate that this inadequate understanding translates to low 'necessity beliefs' towards the administration of chronic medications documented in Africa.[17] We thus further suggest that asthma education programmes should clarify the role of triggers in the precipitation of asthma episodes and the need for medications for asthma control.

### Hesitancy in healthcare workers making an asthma diagnosis resulting in delays and frustration

The delays in children receiving a doctor-diagnosis of asthma identified in our study can possibly be viewed within the context of poor symptom recognition, episodic management and low level of implementation of existing asthma guidelines by healthcare workers especially noted at the PHC level in Africa.[17 26] A systematic review by Mphahlele et al records low rates of asthma diagnosis (38.8%), prescribed medication (47.6%) and an increasing prevalence of severe uncontrolled asthma

in African children.[6] Simplifying diagnostic algorithms for asthma to ensure ease of implementation of care in the African context should be thus viewed as a crucial need. We strongly advocate for this, based on our findings.

### Difficulties experienced coping with adjustments in lifestyles

Children's perspectives on being self-conscious when using inhalers and thus appearing 'different' was an important theme in the present study and has been reported across both resource-developed and challenged countries.[13 19 26] The stigma associated with asthma and the subsequent reluctance to use inhalers in front of peers thus impacts self-management interventions deemed crucial to most asthma care programmes.[19] The use of child-generated strategies where peer interactions help break down social barriers has been suggested for asthma school education programmes.[27] It is relevant to African policy since asthma education programmes must compete with multiple healthcare messaging from common infectious diseases like HIV, tuberculosis and malaria.[20] In this study, we identify peer acceptance as a major barrier in coping with asthma adjustments. We thus recommend that school-going children be sensitised to provide support, specifically with inhaler usage, to affected peers.

Another barrier identified was the difficulty delineating asthma triggers and the inability to alter their environment adequately. Studies from both sSA and elsewhere corroborate this finding, and the use of global checklists that can provide guidance on the range of triggers and viable environment modifications may be useful for inclusion, for example, in the WHO Package of Essential Non-communicable Disease Interventions for Primary Health Care (WHO PEN).[28]

### Difficulties in obtaining hospital care when a child experiences an acute asthma episode

Poor access to asthma care across resource-constrained contexts, including Africa, has been noted as a general barrier and our study corroborates this across all sampled countries.[7] This limited and often difficult access often leads to caregivers seeking alternate avenues for care and delaying appropriate care.[29 30] The WHO PEN attempts to improve asthma management at PHC sites; however, logistical challenges, including lack of essential medications at some sites, require interventions.[28] We specifically identify the need for access to oxygen, nebulisers and monitoring, especially during acute episodes. The concern is that while PHC services may be accessible geographically, the cost of transport, waiting times and loss of income also contribute to poor access. We thus advocate for improved access to both acute and chronic asthma care in PHC settings.

## Easier access to and acceptance of alternate and complementary asthma treatments compared with prescribed asthma medication

The acceptance and usage of traditional medicines and alternative treatments for asthma identified in this study have been previously ascribed to negative perceptions of prescribed medications and seen as inappropriate health-seeking behaviour that contributes to severe asthma.[17 26 30] In the current study, although ambivalence to prescribed medications contributes to poor inhaler use, widespread access to traditional and complementary medicine and indigenous knowledge systems also contributed to alternative treatments. We propose that the propagation of key programmes that advocate for and educate on inhaler therapies and devices should actively engage traditional health systems in the African context to understand better and embed asthma education in these systems.

## Preference for prescribed oral asthma medication with fears and myths over using metred dose inhalers

Multiple studies have corroborated the poor usage and non-adherence to metred dose inhalers in childhood asthma as a major barrier to care.[11 17] This is further seen with the low levels of inhaled corticosteroids (6.7–17%) and 73% of doctor-diagnosed/suspected asthma not using any inhaler documented.[6 31] Over-reliance on short-acting beta agonist inhalers per se is also a barrier to appropriate care.[32] A preference for oral over inhaler-administered medications and widely held concerns that metred dose inhalers were addictive and harmful identified have been noted previously.[10 18] These factors are identified as leading reasons for inadequate use of metred dose inhalers.

With efficacy and safety studies advocating the use of single rapid onset (long-acting beta-agonist/inhaled corticosteroids metred dose inhalers for maintenance, and reliever therapy there is a potential that this single therapeutic option could transform asthma care in sSA.[5 11] Advocacy and policy interventions to make this therapeutic option cheaper need urgent action.[11 17] We suggest that universal access to using a single therapeutic metred dose inhaler, where appropriate, be strongly advocated for in Africa and, in addition, knowledge to dispel fears of using inhaler therapies and devices be included in asthma care plans.

## Strengths and limitations of this study

This study presents a large qualitative data set from multiple sites across Africa, reflecting a wide participant base. The large number of FGD were evenly spread over four countries with the exception of SA having more. The juxtaposition of children with and without a formal diagnosis and their caregivers generated rich data and new insights. We actively sought to identify common barriers across all sites, thus generating data generalisable to the sSA context. Transcription verbatim and using qualitative software to code the data enhanced the reliability. We attempted to ensure a fair and equitable coding team

to decrease in-country biases, and this study thus reflects one of the few unique cross-country analyses from Africa.

Some loss of meaning may have occurred during the translation of African language quotes; however, the data were analysed in the source language by members of the large international coding team who checked the semantic and context equivalence for translated quotes. We did not conduct member checks. Basic demographic data on participants was not provided except for enabling categorisation into the various groups.

We used an inductive approach with constant data comparisons and discussions to inform our understanding of the views expressed. We were aware of the risk of researcher bias in qualitative research and actively discussed reflexivity in the regular multidisciplinary team discussions, which aided analysis and informed interpretation. In addition, we have provided an explicit account for the derivation of the recommendations.

## CONCLUSION

In a continent with rapidly increasing levels of severe uncontrolled asthma, multiple barriers to asthma diagnosis and management are identified along the entire trajectory of care. Exploration of these barriers reveals several broad, generalisable recommendations to modify asthma care plans that can mitigate challenges and potentially transform asthma care in Africa. Advocacy and tailored education are needed to promote access and acceptance for metred dose inhalers, including advocating for access to a single therapeutic, preventative and treatment modality. Healthcare systems require simpler diagnostic pathways and easier emergency access for childhood asthma across the continent.

**Author affiliations**
[1]Paediatrics and Child health, University of KwaZulu-Natal Nelson R Mandela School of Medicine, Durban, South Africa
[2]Department of Paediatrics, King Edward VIII Hospital, Congella, South Africa
[3]Centre for Genomics and Child Health, Queen Mary University of London Faculty of Medicine and Dentistry, London, UK
[4]Child and Adolescent Health Unit, Department of Primary Health Care Sciences, University of Zimbabwe, Harare, Zimbabwe
[5]Department of Psychiatry, Makerere University College of Health Sciences, Kampala, Uganda
[6]Malawi Liverpool Wellcome Programme, College of Medicine, Queen Elizabeth Central Hospital,College of Medicine, Chichiri, Malawi
[7]Lagos State University Teaching Hospital, Lagos State University College of Medicine, Ikeje, Lagos State, Nigeria
[8]Makerere University Lung Institute, Makerere University College of Health Sciences, Kampala, Uganda
[9]Department of Paediatrics and Child Health, Makerere University College of Health Sciences, Kampala, Uganda
[10]Medical School Parirenyatwa Hospital, University of Zimbabwe College of Health Sciences, Harare, Zimbabwe

**Acknowledgements** The authors wish to express our gratitude to all the children, caregivers and patient advisers who participated in this study.

**Contributors** RM, JG, HAM, GM and IT contributed to the study's conception. KLN, REM, SM, ASS, MM, YG, YK, RN and OOA performed the data collection,

checking and coding the transcripts. KLN, REM, SM, ASS, MM, YG, YK, RN and OOA contributed to data analysis and interpretation. KLN, REM, SM and RM drafted the manuscript, and all authors provided critical revisions and editing. All authors reviewed the manuscript. RM and JG are joint guarantors and accept full responsibility for the work, the conduct of the study, had access to the data and controlled the decision to publish.

**Funding** This research was funded by the National Institute for Health Research (NIHR), project number 17/63/38269211, trial registration number: 269211.

**Competing interests** The author(s) declare that they have no financial or personal relationship(s) that may have inappropriately influenced them in writing this article. The funder did not influence the results and compiling of the manuscript. JG reports personal fees from GSK, personal fees from Novartis, personal fees and a grant to QMUL from OM Pharma, personal fees and payment to QMUL from AstraZeneca, personal fees from Omron, outside the submitted work, and is supported by an NIHR Senior Investigator Award. RM reports consultancy and advisory board membership from AstraZeneca, Boehringer and Organon.

**Patient and public involvement** Patients and/or the public were not involved in the design, or conduct, or reporting, or dissemination plans of this research.

**Patient consent for publication** Not applicable.

**Ethics approval** Ethical approval was granted at all the study sites: Malawi, COMREC Reference Number: P.10/18/2494; Nigeria, LREC 06/10/1084; Uganda, MHREC 1514, and UNCST SS 4940; Zimbabwe, MRCZ/A/2415; South Africa, BREC Ref No.: BF002/19. Participants provided written informed consent as well as informed caregiver (parent /guardian) consent and child assent in accordance with both country/local regulations and UK ethics regulations. Participants gave informed consent to participate in the study before taking part.

**Provenance and peer review** Not commissioned; externally peer reviewed.

**Data availability statement** Data are available upon reasonable request.

**ORCID iDs**
Kimesh Loganathan Naidoo http://orcid.org/0000-0003-4940-0534
Sindisiwe Dladla http://orcid.org/0009-0005-4590-9714
Reratilwe Ephenia Mphahlele http://orcid.org/0000-0002-3348-9004
Gioia Mosler http://orcid.org/0000-0002-6900-4080
Sophie Muyemayema http://orcid.org/0009-0002-7514-321X
Andrew Sentoogo Ssemata http://orcid.org/0000-0003-0060-0842
Elizabeth Mkutumula http://orcid.org/0000-0002-4508-6261
Olayinka Olufunke Adeyeye http://orcid.org/0000-0002-1830-0146
Olayinka Goodman http://orcid.org/0000-0003-0930-5989
Yetunde Kuyinu http://orcid.org/0000-0002-8581-9095
Rebecca Nantanda http://orcid.org/0000-0002-5039-8489
Ismail Ticklay http://orcid.org/0000-0002-5925-4692
Hilda Angela Mujuru http://orcid.org/0000-0003-1615-3856
Jonathan Grigg http://orcid.org/0000-0003-3109-6028
Refiloe Masekela http://orcid.org/0000-0001-9665-2035

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
