## [Reviewer comments · BMJ Open]

ARTICLE DETAILS

TITLE (PROVISIONAL)	Barriers to Childhood asthma care in Sub-Saharan Africa: a multi-country qualitative study with children and their caregivers.
AUTHORS	Naidoo, Kimesh; Dladla, Sindiswa; Mphahlele, Reratilwe; Mosler, Gioia; Muyemayema, S; Ssemata, Andrew; Mkutumula, E; Adeyeye, Olayinka; Moyo, Melinda; Goodman, Y; Kuyinu, Y; Nantanda, Rebecca; Ticklay, Ismail; HA, Mujuru; Grigg, Jonathan; Masekela, R

VERSION 1 – REVIEW

REVIEWER	Marangu, Diana University of Nairobi, Paediatrics and Child Health
REVIEW RETURNED	30-Dec-2022

GENERAL COMMENTS	This is a qualitative study conducted in Nigeria, Malawi, Uganda, Zimbabwe and South Africa, among adolescents aged 12-14 years and their caregivers identifying barriers to asthma care. The strengths of the manuscript include: employing a qualitative approach; and multi-country participation. The major weaknesses of the manuscript are:  1. Data Collection  a. In what language/s were the focus group discussions (FGDs) conducted in Nigeria and Uganda? b. What informed the number of FGDs conducted (beyond the four groups described i.e. CAD, PAD, CND, PND)? How did the authors get to 39 focus groups – were all these needed? It is noted that there were 15 FGDs conducted in South Africa compared to four to seven in the other participating countries – may the authors provide an explanation. c. Supplement File 3: Table on FGD 5 indicates teachers participated yet this is not included in the methods section or other parts of the manuscript e.g. abstract/results etc. 2. Results  a. Table 2 – all the major themes and sub-themes seem to be titles rather than themes. e.g. what about understanding asthma? – was there a poor understanding? – this study aims to identify barriers to asthma care, yet the themes do not reflect this. Consider a figure illustrating the relationship of these themes and sub-themes. Would the barriers summarized in Table 3 be more appropriate? b. Some of the thematic headings e.g. reaction to asthma diagnosis do not match with the quotes provided for illustration. c. Did the recommendations provided in Table 4 come from the study participants, or are these extrapolated as potential solutions to the barriers identified in the study? – the two optional questions
---

	in the FGD guide on interventions particularly the first one is not specific to asthma (the answer has not been provided in the results – was there any asthma related health intervention in any of the schools?); and the second question does not appear to be the one asked to come up with the recommendations provided. d. The authors are silent about the teachers FGDs/interviews. 3. Discussion a. The discussion seems to include results (Table 4). b. Was the use of peak flow meters a question that was asked to all study participants? c. “Difficult and limited access to health care during an acute asthmatic episode” – did this come from the study participants? - Were other diagnostic modalities for asthma probed systematically? - if not, it may be inaccurate to suggest that there was no mention. d. The recommendations proposed appear to be those of the authors rather than the study participants – perhaps this needs to be stated explicitly. There are minor textual errors/formatting considerations e.g.  - The term ‘asthmatic children’ – might this be a patient label? (page 4, line 122) - Supplement File 3: Table on FGD 3 – diagnosis of asthma is not numbered; Number 8 – ‘E’ – enablers..
--	---

REVIEWER	Ayuk , Ada University of Nigeria, Paediatrics and Child Health
REVIEW RETURNED	09-Jan-2023

GENERAL COMMENTS	Thank you for the interesting work which is a sub-portion of the bigger quantitative ACACIA work on the burden of asthma in school-going children by assessing asthma control. There is need to further explain the groups studied – for example how was the doctor diagnosis made: was objective measure applied; what constitutes the CND – was it based on symptoms or what exactly. \the methods of statistical analyses and the choice of which excerpts to show in the manuscript is not clear. Its is not clear from the manuscript the questions(apart from the themes) that were discussed; for example the conclusion about advocating for acceptance of PMDI need to be properly highlighted in the methods what was discussed and then the results show more clearly (not just one example) how the responses across the countries showed what the authors deduced. It needs to be convincing enough before it shows in the conclusion as a recommendation. Just main takeaway from this works ought to be clearly highlighted from the very beginning so the the readers could follow the trends accordingly There is also need to clearly justify what extra information/advantage that the qualitative method used brought to the table that was not already done or possible via more objective quantitative methods - these are not clear in the write up Judging from the title of the manuscript these barriers have to be clearly elucidated in the paper and the process of arriving at conclusions and recommendations be clearly linked. The limitations of this study/study methods needs to be clearly put in perspective also
--

REVIEWER	Sharpe, Heather University of Alberta, Medicine
REVIEW RETURNED	27-Mar-2023

GENERAL COMMENTS	Thank you for the opportunity to review this qualitative (thematic analysis) research study outlining the barriers to achieving asthma control for individuals aged 12-14 years of age in five countries in Africa. The manuscript is well-written, the topic is important, and the qualitative methods are well suited to address the research question. I had the following minor suggestions for the authors consideration:  1. The ACACIA Breathing Survey was used to collect information and to identify possible study participants. It would be advantageous to provide some of the results of the survey specific to the study population to have a better understanding of the participants. For example, in table one providing demographic information (identified gender, ethnicity, etc.) would provide a greater understanding of the representativeness of the study participants. Additionally, presenting some information on their experience with asthma (as per section 2 of the survey) would be advantageous to better understand the study participants. 2. The manuscript is a bit long, a common challenge with qualitative research. If necessary it may be feasible to reduce the length to meet the editorial requirements. The authors have used tables and supplementary files effectively to include additional detail. 3. I would suggest using 'children with asthma' rather than asthmatic children. Overall, a well-written thematic analysis on an important research topic.
--

VERSION 1 – AUTHOR RESPONSE

Reviewer 1 Diana Marungu , University of Nairobi	
Data Collection In what language/s were the focus group discussions (FGDs) conducted in Nigeria and Uganda?	The focus group discussions (FGDs) in Nigeria and Uganda were conducted in English. This has been changed and reads : FGDs took place in either English and /or isiZulu (SA), Shona (Zimbabwe) and Chichewa (Malawi). Other centres only used English
What informed the number of FGDs conducted (beyond the four groups described i.e. CAD, PAD, CND, PND)?	The categorisation of the FGDs into those with a formal physician (doctor) diagnosis and those with symptoms but with no formal diagnosis. This was done due to the limited diagnosis of asthma identified in Africa leaving large numbers of children with asthma without a formal diagnosis . ^{5,6,9-12} The four categories follows as we separated parents FGDs and children. With five countries in this research project the aim was to have 1 -2 FGDs purposively selected for each category per country .

	The following change was done in the existing sentence to explicitly indicate the above : Focus groups were conducted face-to-face, with adherence to in-country-specific COVID-19 regulations with at least 1-2 FGD/s per category per country
How did the authors get to 39 focus groups – were all these needed?	The aim was to have 1-2 FGD/s per category per country in order to reflect all groups of both children and their parents . (Those with a formal physician diagnosis and those without a formal diagnosis but with symptoms). The data from multiple focus groups were analysed specifically looking for commonly occurring codes across all countries .
It is noted that there were 15 FGDs conducted in South Africa compared to four to seven in the other participating countries – may the authors provide an explanation.	The plan as described for determining the number of FGDs per category, per country provided a structure to select FGDs. This specific study was situated within a larger ACACIA project and some countries opportunistically sourced a larger number of FGDs than others. Additionally, SA had more FGDs as during the COVID-19 restrictions, numbers in each group were limited in size and there were many parents and children who indicated willingness to participate in these FGDs. This resulted in some countries, especially SA having more FGD than others.
Supplement File 3: Table on FGD 5 indicates teachers participated yet this is not included in the methods section or other parts of the manuscript e.g. abstract/results etc.	The inclusion of the teachers FGDs in Supplement File 3 ,in this manuscript write up was an oversight on our part . This manuscript focuses only on perspectives of children and their caregivers and does not include teachers .The greater ACACIA project did include Teachers however this was not analysed in this study Supplement File 3 has been corrected and the Teachers details removed
Results a. Table 2 – all the major themes and sub-themes seem to be titles rather than themes. e.g. what about understanding asthma? – was there a poor understanding?	The themes and sub-themes have been changed to reflect a greater depth of meanings identified from the data .
this study aims to identify barriers to asthma care, yet the themes do not reflect this. Consider a figure illustrating the relationship of these themes and sub-themes.	With review of the wording of the themes – identified barriers are explicit We have added a figure to show these relationships
Would the barriers summarized in Table 3 be more appropriate?	With the change of the themes , the list of barriers has been changed and hopefully reflect the themes .
b. Some of the thematic headings e.g. reaction to asthma diagnosis do not match with the quotes provided for illustration	We have revised the quotes that were used to decrease the word count and specifically to ensure that only commonly occurring quotes reflecting the themes were included

Did the recommendations provided in Table 4 come from the study participants, or are these extrapolated as potential solutions to the barriers identified in the study?	We have rewritten the process followed with regard to the derivation of the recommendations. This section now reads : Several barriers to asthma care that were common in all five African countries in this study were identified .The relationships of these barriers to optimal asthma care in Africa is highlighted specifically from the participants' perspectives. On further exploration, several cross-cutting recommendations were extrapolated from the analysis of these relationships of the barriers to caring for children with asthma and these were derived from the authors perspectives specifically.
the two optional questions in the FGD guide on interventions particularly the first one is not specific to asthma (the answer has not been provided in the results – was there any asthma related health intervention in any of the schools?);	This manuscript reflected only the data related to childrens and caregivers perspectives. These optional questions were thus not used in the children and caregivers FGD We have removed these optional questions as there were not used and thus not relevant
the second question does not appear to be the one asked to come up with the recommendations provided.	As above this question optional question was not used and was removed .We have explained the derivation of the recommendations
The authors are silent about the teachers FGDs/interviews.	The greater ACACIA study included assessing Teachers perspectives .This Manuscript does not reflect on this data analysis .This manuscript reflects only the caregivers and the children. This manuscript provides the data and analysis of the FGDs of the caregivers and children only
Discussion a. The discussion seems to include results (Table 4).	Table 4 has been removed
Was the use of peak flow meters a question that was asked to all study participants?	Questions on peak flow meters were not asked to study participants
Difficult and limited access to health care during an acute asthmatic episode” – did this come from the study participants? -	The wording has been changed as this was not from the study participants .The wording now reads :difficulties in obtaining hospital care when a child experiences an acute asthma episode.
Were other diagnostic modalities for asthma probed systematically? - if not, it may be inaccurate to suggest that there was no mention.	We have removed the references to diagnostic modalities including peak flow meters
The recommendations proposed appear to be those of the authors rather than the study participants – perhaps this needs to be stated	We have rewritten this section and this now reads: On further exploration, several cross-cutting recommendations were extrapolated from the analysis of these relationships of the barriers to caring for children with

explicitly.	asthma and these were derived from the authors perspectives specifically
There are minor textual errors/formatting considerations e.g. - The term 'asthmatic children' – might this be a patient label? (page 4, line 122)	The term 'asthmatic children' has been changed to children with asthma
- Supplement File 3: Table on FGD 3 – diagnosis of asthma is not numbered; Number 8 – 'E' – enablers..	This has been corrected
Reviewer: 2 Ada Ayuk , University of Nigeria	
There is need to further explain the groups studied – for example how was the doctor diagnosis made: was objective measure applied	We have attempted to explain the derivation of the groups in the study design and methods .As this study was an extension of the greater ACACIA study the details of the disgnostic criteria used has been referenced to the formal protocol of the ACACIA study. Reference 22: Mosler G, Oyenuga V, Addo-Yobo E, et al. Achieving Control of Asthma in Children in Africa (ACACIA): protocol of an observational study of children’s lung health in six sub-Saharan African countries. BMJ Open 2020;10:e035885. doi:10.1136/ bmjopen-2019-035885 Study design The ACACIA project aims to determine the burden of asthma in school-going children by assessing asthma control in children between 12 and 14 years with symptoms compatible with asthma or a doctor-diagnosis of asthma to assess symptom severity and asthma control. As a separate study, we conducted focus group discussions (FGDs) with children with an asthma diagnosis or symptoms suggestive of asthma and their caregivers to obtain their perspectives on the barriers to managing asthma. The study plan, procedures and methods were standardised across all countries.²² Study population Children and their caregivers were recruited initially as part of the ACACIA dataset. A Breathing Survey developed from questions in the Global Asthma Network (GAN) questionnaire (Supplementary file 1) was used to identify participants between 12 and 14 years with asthma symptoms or an asthma diagnosis.²²

what constitutes the CND – was it based on symptoms or what exactly.	This is explained in the study design. The group CND: children with severe wheezing symptoms without a doctor-diagnosis of asthma) Supplementary file 1 was added to provide the information on criteria used Focus group participants (both children and their caregivers) were recruited using purposive sampling and categorised into four groups; children with a doctor-diagnosis of asthma (CAD) and their parents/caregivers (PAD), children with severe wheezing symptoms without a doctor-diagnosis of asthma (CND) and their parents/caregivers (PND), with FGDs held separately for each category. The aim of this group categorisation was to ensure views of both children and caregivers, as well as those with and without a formal asthma diagnosis, were adequately sampled.
the methods of statistical analyses and the choice of which excerpts to show in the manuscript is not clear	The methods section has been changed to clarify the steps in analysis. It now reads : Braun and Clarke’s six-step thematic analysis was then used as the analytic framework to identify the patterns and themes within the coded data.²⁵ Major themes and subthemes were identified based on those commonly occurring codes in all countries .These sub-themes were highlighted and grouped into major themes based on commonly occurring patterns and relationships noted between the identified codes. All investigators reached consensus on these major themes and grouping of sub-themes after multiple iterations. Quotes chosen to illustrate the themes derived from the data were specifically only selected if they resonated across all countries .
Its is not clear from the manuscript the questions(apart from the themes) that were discussed; for example the conclusion about advocating for acceptance of PMDI need to be properly highlighted in the methods what was discussed and then the results show more clearly (not just one example) how the responses across the countries showed what the authors deduced. It needs to be convincing enough before it shows in the conclusion as a recommendation.	We have attempted to clarify the process more explicitly of how we derived the codes ,sub-themes and themes and their relationships .We have also then explicitly provided the process of the derivation of the recommendations derived by the authors . This section now reads : Several barriers to asthma care that were common in all five African countries in this study were identified. The relationships of these barriers to optimal asthma care in Africa is highlighted specifically from the participants’ perspectives. On further exploration, several cross-cutting recommendations were extrapolated from the analysis of these relationships of the barriers to

	caring for children with asthma and these were derived from the authors perspectives specifically.
Just main takeaway from this works ought to be clearly highlighted from the very beginning so the the readers could follow the trends accordingly	We have reviewed the wording of the sub-themes and themes so that they resonate with the identified barriers Refer to the reviewed wording of the sub-themes and themes
There is also need to clearly justify what extra information/advantage that the qualitative method used brought to the table that was not already done or possible via more objective quantitative methods - these are not clear in the write up	We have revised the introduction to highlight this point. It now reads A richer, qualitative understanding of identified barriers to asthma care in African countries will assist with the development of relevant advocacy and policy interventions across the continent.²¹ This study aimed to capture the views of children with asthma with and their caregivers, in sSA ,to identify barriers to asthma diagnosis and care as well as interventions to improve asthma care across countries in Africa.
Judging from the title of the manuscript these barriers have to be clearly elucidated in the paper and the process of arriving at conclusions and recommendations be clearly linked.	We have attempted to clarify this process : Several barriers to asthma care that were common in all five African countries in this study were identified .The relationships of these barriers to optimal asthma care in Africa is highlighted specifically from the participants' perspectives. On further exploration, several cross-cutting recommendations were extrapolated from the analysis of these relationships of the barriers to caring for children with asthma and these were derived from the authors perspectives specifically
The limitations of this study/study methods needs to be clearly put in perspective also	We have added a statement to this section to clarify the limitations : It now reads : We were aware of the risk of researcher bias in qualitative research and actively discussed reflexivity in the regular multidisciplinary team discussions, which aided analysis and informed interpretation. In addition we have provided an explicit account for the derivation of the recommendations.
Reviewer: 3 Dr. Heather Sharpe, University of Alberta	
The ACACIA Breathing Survey was used to collect information and to identify possible study participants. It would be advantageous to provide some of the results of the survey specific to the study population to have a better understanding of the participants. For example, in table one providing demographic	The ACACIA Breathing Survey was used to determine categorisation of the children and their caregivers into the categories of those with a physician diagnosis of asthma and those with symptoms of asthma and not diagnosis. Basic demography of all the ACACIA participants were kept for the initial surveys in the greater study ,however at the time of the focus group discussuions all centres did not correlate basic demography (gender , ethnicity) consistently with those who participated on that day. We

information (identified gender, ethnicity, etc.) would provide a greater understanding of the representativeness of the study participants.	thus could not include this data . We have indicated this as a limitation in the study
, presenting some information on their experience with asthma (as per section 2 of the survey) would be advantageous to better understand the study participants.	Unfortunately we do not have this data on those specific participants in the focus group discussions ..The greater ACACIA study did evaluate some aspects of participants experience with asthma and data on severity has been published .  • Oyenuga V, Mosler G, Addo-Yobo E, et al. 2022: The Assessment of Asthma Control in Diagnosed African Adolescents. Am J Respir Crit Care Med 2022, 205:A2394. • Oyenuga V, Mosler G, Addo-Yobo E et al. 2021: Asthma Symptom Severity and Diagnosis in Adolescents in sub-Saharan Africa. European Respiratory Journal 2021 58: OA2565; doi: 10.1183/13993003.congress-2021.OA2565 However as indicated we did not have specific, isolated data on the FGD participants We have indicated this as limitation.
The manuscript is a bit long, a common challenge with qualitative research. If necessary it may be feasible to reduce the length to meet the editorial requirements. The authors have used tables and supplementary files effectively to include additional detail.	We have attempted to decrease the word count by removing some quotes deemed unnecessary
I would suggest using 'children with asthma' rather than asthmatic children.	This has been changed

VERSION 2 – REVIEW

REVIEWER	Marangu, Diana University of Nairobi, Paediatrics and Child Health
REVIEW RETURNED	15-Jul-2023

GENERAL COMMENTS	Regarding focus group discussions - the authors refer the readers to reference 22 (Mosleh et al), the study protocol: "The study plan, procedures and methods were standardised across all countries.22". The approach to attaining the FGDs in the study protocol differs from the results shown. In the data collection section, the authors indicate: "Focus groups were conducted face-to-face, with adherence to in-country-specific COVID-19 regulations with at least 1-2 FGD/s per category per country."
---

	Although responses have been provided to the reviewers comments, the necessary text to inform the readers has not been included in the manuscript. In the response to reviewers, one of the reasons provided for the 15 FGDs conducted in South Africa compared to four to seven in the other participating countries was the limited group sizes due to COVID restrictions. The other countries also had about 4-7 participants per group. The second reason of participants being more willing seems more likely. It is important to provide these reasons for the readers to understand. Similarly, the reasons provided in the comments to the reviewers are not stated in the manuscript.
--	---

VERSION 2 – AUTHOR RESPONSE

Reviewer 1	
Regarding focus group discussions - the authors refer the readers to reference ²² (Mosler et al), the study protocol: "The study plan, procedures and methods were standardised across all countries.²²". The approach to attaining the FGDs in the study protocol differs from the results shown. In the data collection section, the authors indicate: "Focus groups were conducted face-to-face, with adherence to in-country-specific COVID-19 regulations with at least 1-2 FGD/s per category per country." Although responses have been provided to the reviewers comments, the necessary text to inform the readers has not been included in the manuscript. In the response to reviewers, one of the reasons provided for the 15 FGDs conducted in South Africa compared to four to seven in the other participating countries was the limited group sizes due to COVID restrictions. The other countries also had about 4-7 participants per group. The second reason of participants being more willing seems more likely. It is important to provide these reasons for the readers to understand. Similarly, the reasons provided in the comments to the reviewers are not stated in the manuscript.	The necessary text explaining the reasons for the increased number of groups in South Africa compared with other countries has been added now to the manuscript . This now reads : ‘ South Africa held 15 FGDs compared with 4-6 in all other countries. This was due to more parents and children who indicated willingness to participate in FDGs in SA during the COVID.-19 pandemic than all other countries.’

--	--

1

VERSION 3 – REVIEW

REVIEWER	Marangu, Diana University of Nairobi, Paediatrics and Child Health
REVIEW RETURNED	08-Aug-2023
GENERAL COMMENTS	All comments have been addressed.